Interspecies differences in the transcriptome response of corals to acute heat stress

Da-Anoy Jeric 1 2
Posadas Niño 1 3
Conaco Cecilia cconaco@msi.upd.edu.ph 1
1 Marine Science Institute, University of the Philippines Diliman , Quezon City , Philippines
2 Department of Biology, Boston University , Boston , MA , United States of America
3 Centre for Chromosome Biology, School of Biological and Chemical Sciences, University of Galway , Galway , Ireland
Banaszak Anastazia
Electronic publication date: 2024 Dec 10
Publication date: 2024
Volume: 12
Electronic Location ID: e18627
Received 2024 Aug 26; Accepted 2024 Nov 11
Copyright: ©2024 Da-Anoy et al.
Copyright year: 2024
Copyright holder: Da-Anoy et al.
License: This is an open access article distributed under the terms of the Creative Commons Attribution License, which permits unrestricted use, distribution, reproduction and adaptation in any medium and for any purpose provided that it is properly attributed. For attribution, the original author(s), title, publication source (PeerJ) and either DOI or URL of the article must be cited.
License URL: https://creativecommons.org/licenses/by/4.0/

Keywords: Thermal stress, Adaptation, DNA damage, Seriatopora caliendrum

Funding: Department of Science and Technology Philippine Council for Agriculture, Aquatic and Natural Resources Research and Development (DOST-PCAARRD) Coral Genomics Program This study was funded by the Department of Science and Technology Philippine Council for Agriculture, Aquatic and Natural Resources Research and Development (DOST-PCAARRD) Coral Genomics Program to Cecilia Conaco. The funders had no role in study design, data collection and analysis, decision to publish, or preparation of the manuscript.

==============================
Rising sea surface temperatures threaten the survival of corals worldwide, with coral bleaching events becoming more commonplace. However, different coral species are known to exhibit variable levels of susceptibility to thermal stress. To elucidate genetic mechanisms that may underlie these differences, we compared the gene repertoire of four coral species, Favites colemani, Montipora digitata, Acropora digitifera, and Seriatopora caliendrum, that were previously demonstrated to have differing responses to acute thermal stress. We found that more tolerant species, like F. colemani and M. digitata, possess a greater abundance of antioxidant protein families and chaperones. Under acute thermal stress conditions, only S. caliendrum showed a significant bleaching response, which was accompanied by activation of the DNA damage response network and drastic upregulation of stress response genes (SRGs). This suggests that differences in SRG orthologs, as well as the mechanisms that control SRG expression response, contribute to the ability of corals to maintain stability of physiological functions required to survive shifts in seawater temperature.

Introduction

Coral communities worldwide are increasingly threatened by rising sea surface temperatures due to global climate change (Hughes et al., 2003; Hoegh-Guldberg et al., 2007; Carpenter et al. 2008). These thermal anomalies have escalated both the frequency and severity of coral diseases and bleaching events (Porter et al., 2001; Sutherland, Porter & Torres, 2004), leading to significant shifts in community structure and overall decline in coral cover (Hughes et al., 2003; Hoegh-Guldberg et al., 2007; De’ath et al., 2012). However, stress-tolerant corals may resist local perturbations and climate-change associated stressors, eventually repopulating reefs that have undergone mass bleaching events (Van Woesik & Jordán-Garza, 2011).

Corals exhibit differences in tolerance to thermal stress depending on species (Gibbin et al., 2015), morphology (Alvarez-Filip et al., 2011; McClanahan, Starger & Baker, 2015), or thermal history (Middlebrook, Hoegh-Guldberg & Leggatt, 2008; Wall et al., 2018). For example, pocilloporids generally exhibit a low thermal threshold while an acroporid displays the highest overall tolerance among the Red Sea corals tested in a stress experiment (Evensen et al., 2022). However, thermal tolerance of closely related species (e.g., genus Acropora) can be divergent (Marshall & Baird, 2000; Loya et al, 2001; Hughes et al., 2017; Dalton et al., 2020), which may be attributed to the morphological diversity within a taxonomic group (Baird & Marshall, 2002; Hughes et al., 2017). Massive and encrusting colonies tend to be more resilient to bleaching compared to finely branched species (Loya et al, 2001). In the 1998 bleaching event on Ishigaki Island, massive Porites exhibited higher bleaching resilience than the branching morphotype (Kayanne et al., 2002). This variability in stress tolerance among corals is, in part, attributed to lineage-specific innovations in the molecular toolkit for responding to stress events. For example, comparative genomics of major scleractinian lineages showed that stress tolerance in corals is correlated to the number of genes encoding HSP20 proteins (Ying et al., 2018). A genomic survey of the starlet sea anemone, Nematostella vectensis, revealed that cnidarians have all the components of a typical stress response network, including genes engaged in responding to reactive oxygen species, toxic metals, osmotic shock, thermal stress, pathogen exposure, and wounding (Reitzel et al., 2008). Targeted comparison of the cnidarian stress response gene (SRG) repertoire among diverse coral species within a reef may unveil other determinants of inter-species variability in coral stress tolerance.

Describing the acute and chronic stress response mechanisms in corals reveal short- and long-term acclimatization strategies, which may determine their adaptive capacity. For instance, the acute thermal stress response of coral larvae is accompanied by homeostatic functions (e.g., expression of heat shock proteins), while chronic stress response is associated with homeorhetic regulation (e.g., transcriptome-wide changes and expression shifts of translation machinery) (Meyer, Aglyamova & Matz, 2011). This response was shown to be influenced by stress history, in which frequent exposure to stressful conditions can precondition coral populations, enhancing the thermal tolerance of even the most susceptible groups (Bellantuono et al., 2012; Carilli, Donner & Hartmann, 2012). For example, populations of A. hyacinthus in highly variable environments exhibit thermal tolerance, characterized by elevated constitutive expression, or frontloading, of heat shock proteins, antioxidants, and genes associated with apoptosis, innate immunity, and cell adhesion (Barshis et al., 2013). This adaptation may represent genetically fixed acclimatization responses to recurring variable levels of temperature, pH, and oxygen (Craig, Birkeland & Belliveau, 2001; Smith et al., 2008) that may have persisted over several generations of the local coral population. Elucidating transcriptome-wide changes and expression patterns of SRGs in corals under stress may further uncover molecular mechanisms underlying differences in coral thermal tolerance.

Here, we asked how sympatric coral species in the Bolinao-Anda Reef Complex (BARC) in the northwestern Philippines, a region that experiences thermal anomalies and steadily rising sea surface temperatures (Peñaflor et al., 2009; Yu, 2012), would fare under thermal challenge. Four coral species, Favites colemani (family Merulinidae, superfamily Robusta), Montipora digitata (family Acroporidae, superfamily Complexa), A. digitifera (family Acroporidae, superfamily Complexa), and Seriatopora caliendrum (family Pocilloporidae, superfamily Robusta), were exposed to an experimental thermal regime. We then performed transcriptome sequencing to compare the expression responses of SRG orthologs across species. This genomic information provides a window into the differential susceptibilities of corals to elevated temperature and reveals the molecular mechanisms that may underlie these differences.

Materials and Methods

Coral collection and acclimatization

Three colonies each of F. colemani, M. digitata, A. digitifera, and S. caliendrum (Figs. 1A–1D) were collected from depths of 2–9 m within the Bolinao-Anda Reef Complex (BARC), northwestern Philippines, in November 2016 (Table S1). Sea surface temperatures within the reef range from 25–32 °C with an annual mean temperature of 28.89 ± 0.90 °C based on monitoring data from the Bolinao Marine Laboratory. Samples were collected with the permission of the Philippines Department of Agriculture Bureau of Fisheries and Aquatic Resources (DA-BFAR Gratuitous Permit No. 0102-15). Colonies at least 10–15 m apart were collected to minimize genotypic similarity, although sample genotypes were not evaluated. Corals were fragmented into 2.5–5.0 cm long nubbins (∼10 fragments from each colony per species or ∼30 fragments per species in total) that were reared for two weeks in outdoor tanks to allow healing from the fragmentation process. The tanks were maintained at a seawater temperature of 28 ± 1 °C and illumination under low photosynthetic photon flux density (∼80–90 µmol m−2 s−1) on a 12:12 light-dark cycle. Fragments were tagged to keep track of their colony of origin. Surviving and healed fragments were then allowed to acclimatize for two weeks in indoor experimental tanks with running seawater maintained at 28 ±1 °C and illumination of ∼80 µmol m−2 s−1 on a 12:12 light-dark cycle.

Figure 1 Comparison of coral de novo transcriptomes.

Representative photographs of (A) Favites colemani, (B) Montipora digitata, (C) Acropora digitifera, and (D) Seriatopora caliendrum. (E) Relative abundance of symbiont transcripts with best hits to genomes of other Symbiodiniaceae. Smic, Symbiodinium microadriaticum; Bmin, Breviolum minutum; Cgor, Cladocopium goreaui; Dtre, Durusdinium trenchii; Fkaw, Fugacium kawagutii. (F) Orthologous groups in coral host transcriptomes from this study (in bold font) and in genomes (asterisks) or transcriptomes of other coral species from the Robusta (light blue circle) and Complexa (yellow circle) superfamilies. Family affiliations of corals in this study are indicated by colored diamonds at the nodes (red, Pocilloporidae; blue, Merulinidae; orange, Acroporidae). Pdam, Pocillopora damicornis; Scal, S. caliendrum; Shys, S. hystrix; Lpur, Leptastrea purpurea; Drot, Dipsastrea rotumana; Fcol, F. colemani; Facu, F. acuticollis; Plut, Porites lutea; Aspp, Astreopora spp.; Mdig, M. digitata; Mcac, M. cactus; Aten, A. tenuis; Adig, A. digitifera; Hcoe, Heliopora coerulea. Heatmap colors represent Pearson’s correlation coefficient for orthogroups between species pairs. (G) Number of unique and common PFAM domains identified in the coral host transcriptomes of representative species from four genera. (H) Enriched PFAMs across different coral species.

Thermal stress experiments

Thermal stress experiments were conducted in 40 L tanks supplied with constantly aerated, 10 µm-filtered flow-through seawater, as previously described (Da-Anoy, Cabaitan & Conaco, 2019). Briefly, after two weeks of acclimation, fragments were transferred directly to experimental tanks set at 28 °C (control) or 32 °C (heated). Two independent replicate tanks were used for each temperature treatment. Each treatment tank contained 5–6 fragments from each of the three colonies of the four coral species used in this study (at least 15 coral fragments in total per tank).

Seawater was pumped into treatment tanks from chilled reservoirs at a rate of 5–8 L/h. The temperature in each tank was adjusted to the target setting using submersible thermostat heaters (EHEIM GmbH & Co. KG, Baden Wurttemberg, Germany) with a water pump (600 L/h) to augment circulation. All setups were illuminated under low photosynthetic photon flux density (∼80 µmol m−2 s−1) on a 12:12 light-dark cycle to avoid light stress. Temperature and light intensity were monitored using submersible loggers (HOBO pendant; Onset Computer Corp., Bourne, MA, USA). Coral fragments were collected after 4 and 24 h exposure to treatment conditions. Samples were flash-frozen in liquid nitrogen for transport and then stored at −80 °C before processing.

RNA extraction and sequencing

Total RNA was extracted using TRIzol Reagent (Invitrogen, Waltham, MA, USA) following the manufacturer’s protocol. Contaminating DNA was removed using the DNA-free kit (Invitrogen). RNA integrity of samples was determined by electrophoresis on a native agarose gel with denaturing loading dye. RNA quantification was done using the BioSpec Nanodrop spectrophotometer (Shimadzu, Kyoto, Japan). Samples with an RNA integrity number (RIN) below 7.8 were excluded from sequencing. Three biological replicates from different colonies were collected for control and heated treatments at 4 and 24 h, ensuring each colony was represented in all treatments (12 samples per species). Samples were sent to BGI Genomics (Hong Kong) or Macrogen (South Korea) for library preparation and sequencing. mRNA enrichment and preparation of barcoded cDNA libraries were done using the TruSeq RNA Sample Prep Kit (Illumina, Inc., San Diego, CA, USA). Favites colemani and M. digitata were sequenced on the HiSeq 4000 platform (Illumina, Inc.) at Macrogen, while A. digitifera and S. caliendrum libraries were sequenced on the HiSeq 2500 platform (Illumina, Inc.) at BGI Genomics, to generate 100 bp paired-end reads.

Preprocessing of reads, transcriptome assembly, and annotation

Raw sequence read quality was assessed using FastQC 0.10.1 (Andrews, 2010) and trimmed through Trimmomatic 0.32 (Bolger, Lohse & Usadel, 2014). Poor-quality bases (quality score <3) at leading and trailing bases, as well as the first 15 bases of the reads were discarded. Reads less than 36 bases long and with an average quality per base (4-base sliding window) <30 were trimmed. De novo transcriptome assembly was carried out through Trinity (Grabherr et al., 2011) using eight libraries for each species. The assembled reference transcriptomes were subjected to CD-HIT-EST (Fu et al., 2012) to cluster genes at 90% identity. To further reduce assembly redundancy, transcript abundance estimation was performed by mapping trimmed reads back to the reference transcriptomes using RNASeq by Expectation-Maximization (RSEM) (Li & Dewey, 2011) and the Bowtie alignment method (Langmead et al., 2009). Transcript isoforms with zero isoform percentage (IsoPct) were removed. Isoforms with the highest combined IsoPct or longest length were retained for each transcript. Protein-coding regions for each transcript were predicted using TransDecoder v.2.0.1 (https://github.com/TransDecoder/transdecoder) package in Trinity. Only protein-coding transcripts and their longest predicted peptide sequences were retained for subsequent analyses.

To segregate host and symbiont sequences, Psytrans (https://github.com/sylvainforet/psytrans) was implemented using curated peptide sequence databases of corals (i.e., Montipora capitata (Shumaker et al., 2019), A. digitifera (Shinzato et al., 2011), Goniastrea aspera, Galaxea fascicularis (Ying et al., 2018), A. tenuis, and Porites lutea (ReFuGe 2020 Consortium, 2015) and Symbiodiniaceae representatives (i.e., Fugacium kawagutii, Cladocopium goreaui (Li et al., 2020), Breviolum minutum, Cladocopium sp., Symbiodinium sp. (Shoguchi et al., 2013), and Cladocopium C15 (Robbins et al., 2019), respectively. To further remove potential contaminating sequences from other epibionts, predicted peptides were aligned by Blastp (e-value ≤10−5) against the Genbank non-redundant (nr) sequence database. Only sequences with a best match to Cnidaria (for host) or Dinophyceae (for symbiont) were retained in the final non-redundant reference assemblies. Assembly completeness was assessed through Benchmarking Universal Single-Copy Orthologs (BUSCO) (Simão et al. , 2015) using the Eukaryota, Metazoa, and Alveolata ortholog databases.

The final non-redundant reference transcriptomes were annotated following the Trinotate pipeline (https://trinotate.github.io/). Homolog search was performed by aligning nucleotide and predicted peptide sequences against the UniProt/Swiss-Prot (UniProt, 2019), Genbank RefSeq (O’Leary et al., 2016), and nr databases (Pruitt, Tatusova & Maglott, 2005) through Blastx and Blastp (e-value ≤10−5). Protein domains were identified through HMMER v.3.1b2 (http://hmmer.org) using the Pfam-A database (v31.042). The top Blast hits and identified protein domains for each gene were used as inputs into Trinotate to predict gene ontology (GO) annotations and to generate a comprehensive assembly annotation report.

Ortholog analysis, gene content comparison, and symbiont sequence similarity

Orthologous gene families in the host transcriptomes of F. colemani, M. digitata, A. digitifera, and S. caliendrum and in the genomes or transcriptomes of other coral species were identified using OrthoFinder (Emms & Kelly, 2019). A total of 10 other coral species representing superfamily Robusta (P. damicornis, S. hystrix, Leptastrea purpurea, Dipsastrea rotumana, and F. acuticollis) and superfamily Complexa (Porites lutea, Astreopora spp., M. cactus, and A. tenuis) were included in the analysis, with an octocoral (Heliopora coerulea) as outgroup (Table S2). The classification of these species into the Robusta and Complexa superfamilies was based on the works of Zhang et al. (2019), Okubo (2016) and Ying et al. (2018). Enriched orthologous genes among taxonomic groups were identified using KinFin (Laetsch & Blaxter, 2017). Predicted peptide sequences of representative coral species were annotated against the Pfam 32.0 (Finn et al., 2014) database to annotate expanded orthogroups, as well as to identify lineage-restricted protein domains.

Symbiont transcriptomes from F. colemani, M. digitata, A. digitifera, and S. caliendrum were aligned using Blastp at an e-value cutoff of 1 × 10−5 against genomes of other Symbiodiniaceae representatives, including S. microadriaticum, B. minutum, C. goreaui, Durusdinium trenchii, and F. kawagutii (Table S2). The affiliation of each transcript was assigned based on its top Blastp hits (highest % identity and lowest e-value).

Identification of stress response genes and transcription regulators

Cnidarian stress response genes (SRGs) (Reitzel et al., 2008), as well as gene regulatory elements, including transcription factors (Bahrami, Ehsani & Drabløs, 2015) and epigenetic modifiers (Lee & Workman, 2007; Kooistra & Helin, 2012; Seto & Yoshida, 2014; De Mendoza et al., 2019), were identified in the host transcriptomes of F. colemani, M. digitata, A. digitifera, and S. caliendrum based on their characteristic domains (Tables S3–S4) and top Blastp hit (e-value < 1 × 10−5) against the UniProtKB/Swiss-Prot database.

The abundance of SRGs was computed relative to the total number of predicted peptides in each species. SRGs that distinguish between coral taxonomic groups were identified using the Multiple Variable Associations with Linear Models (MaAsLin 2) (Mallick et al., 2021) package implemented in R.

Differential gene expression analysis

Trimmed reads were mapped back to the concatenated host and symbiont reference transcriptomes to estimate transcript abundance using RSEM (Li & Dewey, 2011) and the Bowtie alignment method (Langmead et al., 2009). Expected counts were converted to counts per million (CPM) and only genes with >2 CPM in at least two libraries (F. colemani = 43,336, M. digitata = 43,320, A. digitifera = 49,343, S. caliendrum = 44,171) were included in differential gene expression analysis. Time-matched pairwise comparisons between control and heated samples were conducted using the edgeR (Robinson, McCarthy & Smyth, 2010) package in R. Transcripts with a log2 fold change (log2FC) ≥ —4— and a false discovery rate (FDR)-adjusted p-value <0.05 were considered differentially expressed. Power analysis using RNASeqPower (Hart et al., 2013) predicts 99% accuracy of detection of true positives at this log2FC given our experimental design and sequencing depth. GO enrichment analysis for differentially expressed genes (DEGs) was performed using the topGO package (Alexa & Rahnenführer, 2009) in R. Only GO terms with p-value <0.01 were considered significantly enriched. Enriched terms were summarized through REVIGO (Supek et al., 2011) at 0.5 similarity cut-off.

Predicted peptides of S. caliendrum were searched against the human proteome v.11.5 from the STRING v.11 database (Von Mering et al., 2003) with an e-value cut-off of 1 × 10−5. Blastp top hits for DEGs in either 4- or 24-h comparisons were used as input in pathway enrichment analysis. Protein–protein interactions of genes involved in enriched pathways (score > 0.400 and FDR < 0.01) were retrieved from the STRING v.11 database (Mering et al., 2003). Interaction networks were visualized using Cytoscape v.3.7.2 (Shannon et al., 2003). Relative expression of S. caliendrum gene homologs in heated samples relative to the controls was computed as the average sum of transcripts per million (TPM).

To assess the effect of treatments, raw counts of host- and symbiont-derived transcriptomes were rlog-transformed and used as input for principal component analysis (PCA) with plotPCA (DESeq2 package), followed by PERMANOVA using the adonis2 function from the vegan package (Oksanen et al., 2022). Gene expression plasticity in response to treatments was calculated as the distance between an individual’s transcriptome profile and the mean of all samples in the 4 h control group (Bove et al., 2023). Differences in plasticity between treatments were tested using an ANOVA followed by Tukey’s HSD post-hoc tests.

Visualization

All figures were generated using the ggplot2 package (Wickham, 2016) in R. Phylogenetic trees were visualized in iTOL (Letunic & Bork, 2019).

Results

De novo assembly of four scleractinian coral transcriptomes

High-throughput sequencing of F. colemani, M. digitata, A. digitifera, and S. caliendrum (Figs. 1A–1D) transcriptomes generated 368.74 to 688.08 M raw reads per species (Table S5). Quality-filtered reads were assembled de novo (Table S1). Resulting assemblies were subjected to sequence similarity clustering, isoform selection, and removal of non-protein-coding and non-Cnidaria or non-Dinophyceae sequences to generate non-redundant reference transcriptomes for each species (F. colemani: n = 52,832, Ex90N50 = 2,123; M. digitata: n = 51,324, Ex90N50 = 2,045; A. digitifera: n = 65,543, Ex90N50 = 2,379; S. caliendrum: n = 54,146, Ex90N50 = 2,346) (Table 1, Table S1). Assembly statistics are comparable across species sequenced on different platforms (Table S1, Fig. S2–S4). Assemblies contained a greater proportion of symbiont (48.25–55.93%, GC content = 54.94–56.04%) compared to host transcripts (44.07–51.75%, GC content = 41.78–42.82%) (Table 1, Table S1). BUSCO analysis revealed that the host assemblies were about 90.90–93.80% complete for metazoan core genes and the symbiont assemblies were 66.70–70.80% complete for alveolate core genes (Table 1, Table S6). Around 67.08–73.41% of host transcripts were annotated by at least one of the following databases: UniProtKB/Swiss-Prot, PFAM, or GO. Symbiont assemblies had a relatively lower annotation rate of 54.92–57.90% (Table S7).

Table 1 Assembly statistics of de novo transcriptomes of four scleractinian corals.

	Favites colemani	Montipora digitata	Acropora digitifera	Seriatopora caliendrum	
Superfamily	Robusta	Complexa	Complexa	Robusta	
Family	Merulinidae	Acroporidae	Acroporidae	Pocilloporidae	
Total transcripts	52,832	51,324	65,543	54,146	
Total transcripts ExN50	2,123	2,045	2,379	2,346	
Host bin					
Number of Transcripts	25,681	22,616	33,918	24,763	
G+C content (%)	42.82	42.51	42.03	41.78	
% completeness (BUSCO Metazoa db)	92.00	93.00	93.80	90.90	
Symbiont bin					
Number of transcripts	27,151	28,708	31,625	29,383	
G+C content (%)	55.03	55.24	54.94	56.04	
% completeness (BUSCO Alveolata db)	70.80	69.00	67.30	66.70	

Symbiont transcript affiliations

Symbiont assemblies were aligned by Blastp to genomes of other Symbiodiniaceae to determine the possible identities of microalgal symbionts associated with the four coral hosts. About 51.07–55.26% of symbiont transcripts in F. colemani, M. digitata, and A. digitifera had a best hit with sequences from C. goreaui and 6.84–7.45% to D. trenchii (Fig. 1E, Table S8). In contrast, only 9.10% of symbiont transcripts in S. caliendrum had a best hit with C. goreaui, while 67.69% showed highest similarity with sequences from D. trenchii.

Comparison of coral host assemblies

Ortholog analysis was conducted to assess gene conservation across coral species and to explore possible gene expansion events. The gene repertoire of our coral host assemblies was comparable to other coral transcriptomes or genomes, indicating that we were able to capture most of the scleractinian core genes. Correlation of orthologous groups that were identified across species recapitulated phylogenetic groupings, with S. caliendrum and F. colemani clustering with other members of suborder Vacatina (Robusta superfamily) and M. digitata and A. digitifera clustering with other members of suborder Refertina (Complexa superfamily) (Fig. 1F, Table S9). The majority of transcripts from our assemblies are conserved in Anthozoa (4,296 orthogroups) and scleractinian representatives (740 orthogroups) (Table S10). 43 orthogroups are represented in Robusta and 17 in Complexa. Family-specific orthologous genes were also identified among representatives of pocilloporids (n = 57), merulinids (n = 44), and acroporids (n = 47).

Ortholog-based and taxon-aware analyses using KinFin (Laetsch & Blaxter, 2017) revealed genes that had undergone lineage-specific expansions (Table S11). An ion transport protein (OG224) and secretin GPCR (OG190) are relatively enriched in members of family Pocilloporidae, while HSP70 (OG128), NHL repeat (OG141 and OG236), B-box zinc finger (OG134), and carboxylesterases (OG233) are expanded in merulinid species. A total of 30 orthogroups are enriched among acroporids, including rhodopsin GPCR (OG254), ubiquitin transferase (OG178), DNA-binding THAP (OG13), reverse transcriptases (OG38, OG0, and OG205), and helicases (OG145 and OG179), as well as a diversity of integrase (OG10), transposases (OG229, OG71, OG49, OG78, OG73, and OG123), and endonucleases (OG44, OG230, OG222, OG110, and OG281). Immune-related orthogroups, such as NOD-like receptors (OG29), immunoglobulin (OG76 and OG156), and TRAF-type zinc finger (OG274), were also enriched among members of genus Acropora.

Gene orthologs and SRG repertoire

To evaluate the functional genes represented in selected coral genera, we compared protein domains identified in the predicted peptides of Favites (F. colemani and F. acuticollis), Montipora (M. digitata and M. cactus), Acropora (A. digitifera and A. tenuis), and Seriatopora (S. caliendrum and S. hystrix). This revealed 4,728 domains conserved in all four genera. A total of 163 protein domains were shared between Favites, Montipora and Acropora (n = 163), including an ABC transporter family (PF06541), flavoprotein (PF02441), DNA damage repair protein (PF08599), PAC3 (PF10178), Pcc1 (PF09341) and heat shock transcription factor (PF06546) (Fig. 1G, Table S12). Domains restricted to Seriatopora and Favites (n = 44) included players involved in fungal-like histidine biosynthetic pathway (i.e., HisG (PF01634), HisG_C (PF08029), Histidinol_dh (PF00815), IGPD (PF00475), PRA-CH (PF01502), and PRA-PH (PF01503)), as well as ectoine synthase (PF06339), selenoprotein P (PF04592), an ER-bound oxygenase (PF09995), FOXO-TAD (PF16676), and transmembrane protein families (PF3616, PF07114, PF16070, and PF15475) (Table S12). On the other hand, protein domains specific to Montipora and Acropora (n = 77) included cell cycle regulatory protein Spy1 (PF11357), serpentine type GPCR (PF10318), DNA binding proteins (HTH_32 (PF13565), MBF2 (PF15868), TBPIP (PF07106), and Tfb5 (PF06331), cutA1 divalent ion tolerance protein (PF03091), membrane transport protein (PF03547), HSF binding factor (PF06825), stress-tolerance associated domain CYSTM (PF12734), Sep15/SelM redox domain (PF08806), zinc fingers (PF02892 and PF11781), and protein domains linked to DNA damage response and repair (UPF0544 (PF15749) and TAN (PF11640)). A total of 99 domains were restricted to Seriatopora (e.g., E3 ubiquitin ligases (PF12185 and PF10302), ROS modulator Romo1 (PF10247), odorant receptors (PF02949), and NF-κB modulator NEMO (PF11577)), 75 to Favites (e.g., C2H2 type zinc finger (PF13909), oxidoreductase (PF07914), and serpentine chemoreceptor (PF07914)), 44 to Montipora (e.g., serpentine chemoreceptor (PF10292)), and 93 to Acropora (e.g., cell wall stress sensor (PF04478) and peroxidase (PF00141)).

To gain insights into the stress response gene (SRG) repertoire of each coral species, we identified known cnidarian SRGs involved in chemical, pathogen, and wounding stress (Reitzel et al., 2008). Most SRG families were represented in each species (Table S13). Proteins engaged in chemical stress response (i.e., HSP90, phytochelatin, and Fe/Mn SOD) and pathogen defense (i.e., glycosyl hydrolase) were enriched in members of superfamily Robusta, particular in the merulinids, while proteins with flavin-binding monooxygenase and caspase recruitment domains were relatively more abundant in members of superfamily Complexa (Fig. 1H, Tables S14–15).

Transcriptome response to elevated temperature

To determine how the different coral species respond to the same thermal stress regime, we subjected coral fragments to acute elevated temperature (32 °C vs 28 °C) for 4 and 24 h. Principal component analysis showed shifts in global gene expression profiles between heated and control samples for host-derived transcripts in A. digitifera, F. colemani, and S. caliendrum (Figs. 2A–2D), as well as for symbiont-derived transcripts in M. digitata and S. caliendrum (Figs. S1A–S1D). These changes were quantified using gene expression plasticity analysis, which showed a significant difference in treatments only in S. caliendrum (p-value ≤0.05), indicating significantly higher plasticity in heated treatments, particularly at 24 h exposure (Figs. 2A–2D).

Figure 2 Transcriptome dynamics under thermal stress.

Principal component analysis (PCA) and gene expression plasticity plots of host-derived transcriptomes for (A) F. colemani, (B) M. digitata, (C) A. digitifera, and (D) S. caliendrum in different treatments. The x- and y-axes represent the % variance explained by the first two principal components. (E) Differentially expressed (log2FC ≥ —4—, FDR-adjusted p-value < 0.05) host and symbiont genes (upregulated, top; downregulated, bottom) after 4 h and 24 h exposure.

Differential expression analysis supports this finding, with fewer genes exhibiting a significant change in expression (log2FC ≥ —4—, FDR ≤ 0.05) in F. colemani (94 at 4 h, 2 at 24 h), M. digitata (2 at 4 h, 0 at 24 h), and A. digitifera (124 at 4 h, 1 at 24 h) subjected to elevated temperature (Fig. 2E, Table S16). In contrast, S. caliendrum had 2,865 differentially expressed transcripts at 4 h (2,495 upregulated, 370 downregulated), and 823 transcripts differentially expressed at 24 h (559 upregulated, 263 downregulated). Relative to time-matched controls, more transcripts were differentially expressed at 4 h compared to the 24 h timepoint in all species. Majority of differentially expressed transcripts originated from the coral host.

Host genes upregulated in F. colemani subjected to acute thermal stress included antioxidants (MOXD1 and PXDNL), solute carrier proteins (S6A13, SO4A1, and COPT2), and ABC transporters (2 MRP4), as well as ECM components (HMCN2, 2 SNED1, SUSD2, and MYO1), whereas heat shock proteins (HSP7C and HSP16) and ubiquitination-related genes (TRI50 and UBC12) were downregulated (Table S17). In A. digitifera, oxidoreductase (QORL1) and ECM-associated genes (ITAD and MDGA2) were upregulated, whereas HSP7A, E3 ubiquitin protein ligases (R113A and TRAF6), and translation initiation factors (2 IF4G1) were downregulated (Table S18).

Seriatopora caliendrum thermal stress response

In contrast to the other corals that showed a minimal transcriptional response to the experimental treatment, S. caliendrum exhibited a significant shift in expression of genes involved in diverse biological processes, including basic cellular functions (e.g., growth, cell cycle, cell communication, cell adhesion, and DNA replication), metabolism (e.g., glyoxylate cycle, carbohydrate metabolism, nitrogen metabolism, DNA biosynthesis, cholesterol catabolism, and xenobiotic metabolism), gene expression control (e.g., microRNA-mediated gene silencing, DNA methylation, and post-translational modification), immune response (e.g., endocytosis, bacterial agglutination, and interferon-beta production), and stress response pathways (e.g., mismatch repair, heat acclimation, response to decreased oxygen levels, oxygen radical, and pH) (Tables S19–20). These broadscale transcriptional changes were accompanied by dynamic expression of gene regulatory elements such as epigenetic modifiers (DNA methyltransferases (DNMT1 and DNM3A), thymine DNA glycosylase (UNG and TDG), histone deacetylase (HDAC6), methyltransferases (PRD14, PRDM6, and SMYD3), and demethylases (KDM8, JMJD4, and KDM1B)), along with diverse transcription factors (ARID, bZIP, E2F, Ets, Forkhead, GATA, H2TH, bHLH, HMG box, Homeobox, Myb, Pou, T-box, THAP, and zinc finger factors) (Fig. 3A, Table S21). The dynamic expression of these regulators signals active involvement of transcriptional control in the thermal stress response of S. caliendrum. Most of these regulators exhibited notable change in expression at 4 h and returned to their basal levels after 24 h (Fig. 3A), coinciding with observed transcriptome-wide dynamics (Figs. 2A–2E).

Figure 3 Transcriptional response of S. caliendrum to thermal stress.

(A) Differentially expressed members of gene regulatory families at 4 h (yellow triangles) and 24 h (orange circles). (B) KEGG pathways enriched in the set of differentially expressed genes at 4 h or 24 h timepoints. Gene counts (x-axis) and FDR-adjusted p-values (bubble size) are shown. (C) DNA damage response network in S. caliendrum at 4 h exposure. Relative expression of genes was computed as the sum of TPM values relative to time-matched control samples. The network is based on human protein–protein interactions.

Activation of the DNA damage response in S. caliendrum

Pathway enrichment analysis of differentially regulated genes in S. caliendrum revealed links to cellular responses to DNA damage (Fig. 3B, Tables S22–23). Reconstruction of the protein interaction network for genes related to the DNA damage response in S. caliendrum revealed upregulation of DNA damage sensors, ATM and ATR kinases, along with CHEK2 kinase (Fig. 3C, Table S24), which set off checkpoint-mediated cell cycle arrest, DNA repair activation, and apoptosis via the p53 pathway (Blackford & Jackson, 2017). Negative regulators of entry into S-phase (RB1) and M-phase (WEE1 and PKMYT1) were also upregulated, further indicating cell cycle arrest despite downregulation of the growth arrest and DNA damage-inducible protein (GADD45A) and upregulation of cyclin-dependent kinases (CDK1/2), cyclin regulatory subunits (CCND2, CCNE1, CCNA1, CCNB2/3), and E2F transcription factor (E2F5). Cell cycle arrest may facilitate activity of DNA repair mechanisms, as evidenced by upregulation of the mediator of DNA damage checkpoint protein 1 (MDC1) (Abraham, 2001).

There was also increased expression of genes implicated in DNA damage surveillance and removal, including players in mismatch repair, base excision repair, nucleotide excision repair, homologous recombination, alternative end-joining, and trans-lesion synthesis (Fig. 3C, Table S24) (Li & , 2008; Hakem et al., 2012; Bienko et al., 2010; Krokan & Bjøås, 2013). This was accompanied by activation of DNA helicases (MCM2/4/5/6), DNA polymerase subunits (POLD3, POLE, POLE2/3/4, POLH), proliferating cell nuclear antigen (PCNA), DNA ligase (LIG1), and flap endonuclease (FEN1), which ensure high-fidelity DNA re-synthesis and ligation (Timson, Singleton & Wigley, 2000). Activation of the Fanconi Anemia pathway, which coordinates classical DNA repair pathways (Moldovan & D’Andrea, 2009), suggests a well-orchestrated deployment of DNA repair mechanisms in S. caliendrum under stress.

High levels of DNA damage block mitotic exit through the spindle assembly checkpoint (SAC) (Mikhailov, Cole & Rieder, 2002). The DNA damage network of S. caliendrum (Fig. 3C) revealed downregulation of separin (ESPL1) and activation of SAC components, including the mitotic checkpoint serine/threonine-protein kinase (BUB1) and dual specificity protein kinase (TTK), which inhibit the anaphase promoting complex (ANAPC1/4/5 and CDC23) (Overlack, Krenn & Musacchio, 2014). Increased expression of p53-induced death domain-containing protein 1 (PIDD1) and an executioner caspase (CASP3), along with associated adapter proteins (CRADD and RIPK1), signals activation of apoptosis, which is likely if DNA lesions remain unrepaired.

Expression patterns of stress response genes across species

Comparison of SRG transcript abundance revealed higher basal expression in F. colemani, M. digitata, and A. digitifera relative to the median expression of all transcripts in each species (Fig. 4A). Around 53–64% of SRGs in these corals may be considered frontloaded (1,745 in F. colemani, 1,835 in M. digitata, and 2,063 in A. digitifera). Notably, SRG levels in these species remained stable even under acute thermal stress, with average —log2FC— ranging from 0.21 to 0.58 (Fig. 4B, Tables S25–27). These frontloaded genes include heat shock proteins (HSP70) and efflux pumps (ABC transporters and ion transport proteins), as well as SRGs engaged in redox (aldehyde dehydrogenases, cytochrome p450, aldo/keto reductases, peroxidases, thioredoxins, and glutaredoxins) and conjugative (glutathione S-transferases and sulfotransferases) biotransformation (Fig. 4C). Ferritin and apolipoproteins, which have been shown to correlate with enhanced oxidative stress tolerance (Fischer et al., 2013; Granados-Cifuentes et al., 2013), are among the most highly expressed SRGs in F. colemani (FRIS and VIT), M. digitata (2 FRIS, APLP, and APOB), and A. digitifera (2 FRIS and VIT) (Tables S25–27). Immune receptors (GPCRs, LDL receptors, SRCRs, immunoglobulins, NACHT-, LRR-, Death-, CARD- and TIR-containing proteins) and wound healing genes, such as ECM components (cadherins, collagen, fibronectins, laminins, thrombospondins, and von Willebrand factors), signaling molecules and receptors (activins, galactose binding lectins, C-type lectins, TGF beta, and TNFs), and MH2 transcriptional regulators, also showed stable high expression under the tested conditions (Table S28). It is worth noting that only 27% of SRGs (1,164 genes), including chemical stress response gene families, are frontloaded in S. caliendrum (Figs. 4B–4C, Tables S28–29).

Figure 4 Expression patterns of stress response genes.

(A) Expression distribution of all genes at 28 °C (gray shaded area), SRGs at 28 °C (blue line) and 32 °C (red line). Vertical lines indicate median values for each distribution. (B) Expression dynamics of SRGs showing constitutive expression at 28 °C (y-axis) and fold change under thermal stress (x-axis). Y-intercepts indicate median values and numbers above and below this line denote SRGs with higher or lower constitutive expression, respectively, relative to the median. Points to the left and right of the x-intercept represent genes that are down- and upregulated, respectively. (C) Frontloaded chemical stress response genes. Only genes with normalized counts higher than the transcriptome median expression and with log2FC < —2— in the heated treatment were considered frontloaded. Bubble size indicates the number of frontloaded genes and asterisks denote enrichment relative to the total number of peptides in each species. (D) Average fold change of genes with low basal expression in S. caliendrum that are upregulated under thermal stress. Only chemical SRGs are shown.

Unlike in the other corals, SRGs in S. caliendrum showed a more dynamic shift in expression under heat stress (ave. —log2FC—: 4 h = 1.70, 24 h = 1.15) (Figs. 4A–4B, Table S29). This was even more apparent for SRGs with basal expression lower than the median for the transcriptome (n = 1,637, 47.42%). These genes showed a greater shift in expression (ave. —log2FC—: 4 h = 2.69, 24 h = 1.87) (Fig. 4B). Other genes (n = 524) showed higher average —log2FC— at both 4 h (ave. log2FC = 4.42) and 24 h (ave. log2FC = 3.39) timepoints (Table S30). These lowly expressed but stress-responsive S. caliendrum genes are comprised of protein families that are typically frontloaded in F. colemani, M. digitata, and A. digitifera (Fig. 4D, Table S30). Other SRGs in S. caliendrum (HSP20, organic anion transporter polypeptides, multicopper oxidases, phytochelatin, transferrin, UDP-glucoronosyl and UDP-glucosyl transferase), cell adhesion (fibroblast growth factors, fibronectins, hemopexin, integrins, and nidogen-like), and innate immunity (inhibitor of apoptosis domain, DEAD, lipoxygenases) proteins also exhibited similar expression patterns (Fig. 4D, Table S30).

Discussion

In this study, we sought to understand the molecular mechanisms underlying differences in the physiological response of corals to a common thermal stress regime. Using transcriptome sequencing, we identified gene expression signatures of stress response in these corals. We discovered that gene expression responses, particularly for known stress response genes (SRGs), varied across species, indicating that different corals have distinct strategies to combat thermal stress.

Variation in SRG copies among corals

Comprehensive analysis of genomic and transcriptomic data from diverse coral species (Bhattacharya et al., 2016; Zhang et al., 2019) have revealed the presence of common stress-related pathways. These previous studies and our current work also show that different corals possess varying numbers of stress-related genes, which suggests that certain species are equipped with a more diverse set of SRGs (Cunning et al., 2018; Shumaker et al., 2019; Ying et al., 2018; Voolstra et al., 2017; Robbins et al., 2019). Gene family expansion often gives organisms an adaptive advantage as the presence of multiple gene copies may allow for functional diversification or for generation of more gene products (Zhang et al., 2019; Dougan et al., 2024). Expansion of gene families involved in cellular signaling, stress response pathways, and immunity has been reported in the genomes of certain species, including Pocillopora acuta, Stylophora pistillata, and A. digitifera (Cunning et al., 2018; Voolstra et al., 2017). The presence of multiple copies of innate immunity genes could support greater specificity of microalgal endosymbiont recognition (Emery, Dimos & Mydlarz, 2021; Noel et al., 2023) and the ability to recognize and mount responses against pathogens (Shinzato et al., 2011; Baumgarten et al., 2015; Gittins et al., 2015; Alderdice et al., 2022). Higher copy numbers of heat shock proteins (Ying et al., 2018) and fluorescent proteins (Dizon et al., 2021) in massive corals, such as F. colemani, could help them better respond to environmental stressors, thereby increasing their chances of survival and reproduction during bleaching events or under thermal and acidification stress (Marshall & Baird, 2000; Da-Anoy, Cabaitan & Conaco, 2019; Tañedo et al., 2021).

Transcriptome plasticity and frontloading of SRGs as an adaptive strategy

The capability of the coral host to modify gene expression in response to environmental stress is critical for recovery and survival (Franssen et al., 2011; Seneca & Palumbi, 2015). Global transcriptome change or transcriptome plasticity is a mechanism that promotes the activation of pathways that mediate protection of cellular components or repair of cellular damage (Kenkel et al., 2016; Studivan, Milstein & Voss, 2019; Castillo et al., 2024; Drury et al., 2022; Armstrong et al., 2023). Corals that thrive in highly variable environments typically exhibit greater transcriptome plasticity compared to corals from more stable environments (Kenkel et al., 2016). One of the most notable observations in our study was that the corals showed different responses to acute thermal stress exposure. While F. colemani, M. digitata, and A. digitifera exhibited no visible bleaching and little change in gene expression, S. caliendrum showed a rapid shift in gene expression profile prior to the onset of bleaching (Da-Anoy, Cabaitan & Conaco, 2019). This change in expression after just 4 h of exposure reflects rapid activation of the stress response toolkit in S. caliendrum. However, about 24% of these differentially expressed host genes remained differentially regulated relative to controls at 24 h of exposure, indicating a low level of gene recovery. At this point, the coral may have nearly exceeded its thermal response limit and exhausted its cellular resources, which could explain the bleaching observed upon further exposure (Da-Anoy, Cabaitan & Conaco, 2019). A similar response has been reported in the corals, S. pistillata (Savary et al., 2021) and A. hyacinthus (Thomas et al., 2019), where failure to return to baseline levels of gene expression resulted in low survival, likely due to depletion of energy resources.

It should be noted that, across all species examined, the transcriptional response to thermal stress was more pronounced in the coral host than in the algal symbionts. This aligns with previous reports suggesting that the host may insulate its symbionts from external stressors (Barshis et al., 2014; Davies et al., 2018; Kaniewska et al., 2015). However, we expect that more gradual or longer durations of exposure may elicit different responses than what we observed in the acute thermal stress regime used in this study.

The susceptibility of S. caliendrum to thermal stress contrasts with the broader understanding that transcriptome plasticity often enhances coral resilience (Kenkel et al., 2016; Rivera et al., 2021). Although rapid gene expression changes indicate a high degree of plasticity, these responses were insufficient to prevent bleaching. This suggests that while S. caliendrum can activate protective gene pathways, the rapid onset of these changes may also signal an impending threshold of physiological tolerance. The inability to recover baseline levels of gene expression could deplete cellular resources, which may explain the susceptibility of this species to thermal stress. These findings indicate that transcriptome plasticity, though often beneficial, may have a limited effect especially in corals subjected to extreme or prolonged environmental stressors (Savary et al., 2021; Thomas et al., 2019; Barshis et al., 2014).

In contrast to S. caliendrum, F. colemani, M. digitata, and A. digitifera exhibited greater tolerance to acute thermal stress. These species showed higher baseline expression of SRGs at ambient temperature and the expression of these genes remained stable even at elevated temperature. This provides evidence for transcript frontloading, an adaptive molecular response where protective genes are expressed by the cell at higher levels in anticipation of possible stress exposure (Barshis et al., 2013). For example, corals living in warmer or more variable conditions constitutively upregulate a set of genes that are usually only expressed during heat stress (Barshis et al., 2013; Fifer et al., 2021). Frontloading of SRGs is a likely adaptation for corals in habitats that frequently experience stressful conditions and could support tolerance to thermal fluctuations (Bay & Palumbi, 2017; Castillo & Helmuth, 2005; Kenkel et al., 2016; Mayfield, Fan & Chen, 2013; Seneca & Palumbi, 2015).

DNA damage repair in a thermally sensitive coral

Functional analysis of differentially expressed genes in S. caliendrum revealed evidence for upregulation of protein degradation, transport, DNA damage repair, homeostasis, detoxification, and catabolic processes, which are associated with mechanisms involved in maintaining stable conditions within the coral holobiont. This emphasizes the importance of removing damaged molecules and maintenance of protein conformation and activity (Kenkel et al., 2011; Leggat et al., 2011; Meyer, Aglyamova & Matz, 2011; Rodriguez-Lanetty, Harri & Hoegh-Guldberg, 2009; Rosic et al., 2011) to properly regulate cellular processes that would then allow the recovery and survival of the organism (Traylor-Knowles et al., 2017).

Maintenance of DNA integrity is essential for homeostasis and survival both in stable and stressful environments (Giglia-Mari, Zotter & Vermeulen, 2011). DNA damage is a consequence of high oxidative stress through the generation of reactive oxygen species (ROS) that results in DNA, protein, and lipid damage (Lesser & Farrell, 2004; Lesser, 2005) and dysbiosis between coral host and algae (Inoue & Kawanishi, 1995; Keyer, Gort & Imlay, 1995; Keyer & Imlay, 1996). It is possible that thermal susceptibility of S. caliendrum is, in part, due to lower constitutive expression of antioxidant systems, which may be unable to keep DNA damage at levels that can be managed by expressed repair mechanisms. These findings indicate that the coral is capable of mobilizing cellular mechanisms to regain homeostasis and could possibly survive brief periods of acute stress. It is therefore warranted to examine whether these cellular mechanisms would support the survival of thermally sensitive corals under gradual or periodic thermal stress events.

Conclusion

This study contributes valuable genetic information on four common coral species in the Indo-Pacific. We show that these corals possess all the typical genes required to mount an appropriate stress response and are able to express these, especially during stress conditions. Notably, the corals showed very different responses to the same thermal regime. Species that were resistant to thermal stress showed signs of readiness in terms of gene repertoire, with expanded SRG families, as well as in terms of frontloading, with SRGs constitutively expressed in anticipation of stress. The combination of possessing these genetic toolkits and being able to regulate expression in response to various stress events can spell the difference between survival and bleaching. This variation in adaptive strategies could reflect differences in abundance and distribution of these corals on reefs that experience different thermal fluctuations. Future studies to evaluate the potential of these genes and expression patterns as biomarkers for coral thermotolerance will help us better understand the impact of thermal stress across different locations.

Supplemental Information

Supplemental Information 1 Symbiont transcriptome dynamics under thermal stress

Principal component analysis (PCA) and gene expression plasticity plots of symbiont-derived transcriptome profiles for (A) A. digitifera, (B) F. colemani, (C) M. digitata, and (D) S. caliendrum in different treatments. The x- and y-axes represent the % variance explained by the first two principal components.

Supplemental Information 2 Read count mapping statistics

Number of raw sequence reads per library for each coral species. Acropora digitifera and Seriatopora caliendrum were sequenced on HiSeq2500, while Montipora digitata and Favites colemani were sequenced on HiSeq4000.

Supplemental Information 3 PCA of SRG orthologs present in the 4 coral species

PERMANOVA revealed a significant difference based on species but no significant difference based on sequencing platform.

Supplemental Information 4 Rarefaction curves showing number of assembled transcripts versus read counts

Acropora digitifera (A) and Seriatopora caliendrum (C) were sequenced on HiSeq2500, while Montipora digitata (B) and Favites colemani (D) were sequenced on HiSeq4000. Most libraries exhibit a plateau in the number of detected transcripts indicating sufficient sequencing depth. Differences in number of transcripts across holobionts (coral host and symbionts) are readily apparent, however, these are independent of the sequencing platform used and are likely species-specific signatures.

Supplemental Information 5 Supplemental Tables

Additional Information and Declarations

Competing Interests

Author Contributions

Field Study Permissions

Data Availability

The authors declare there are no competing interests.

Jeric Da-Anoy conceived and designed the experiments, performed the experiments, analyzed the data, prepared figures and/or tables, authored or reviewed drafts of the article, and approved the final draft.

Niño Posadas analyzed the data, prepared figures and/or tables, authored or reviewed drafts of the article, and approved the final draft.

Cecilia Conaco conceived and designed the experiments, analyzed the data, prepared figures and/or tables, authored or reviewed drafts of the article, and approved the final draft.

The following information was supplied relating to field study approvals (i.e., approving body and any reference numbers):

Samples were collected with the permission of the Philippines Department of Agriculture Bureau of Fisheries and Aquatic Resources (DA-BFAR Gratuitous Permit No. 0102-15).

The following information was supplied regarding the deposition of DNA sequences:

De novo transcriptome assemblies used in this study are deposited at DDBJ/EMBL/GenBank under the following accessions: GIVN00000000 (F. colemani), GIVM00000000 (M. digitata), GIVI00000000 (A. digitifera), and GIVG00000000 (S. caliendrum).

Sequence reads are available in NCBI under the following BioProject accessions: PRJNA422022 (F. colemani), PRJNA422015 (M. digitata), PRJNA421253 (A. digitifera), and PRJNA422012 (S. caliendrum).

The final non-redundant reference transcriptomes, as well as datasets generated and analyzed in this study, are available at Figshare: Posadas, Niño; Da-Anoy, Jeric; Conaco, Cecilia (2024). Stress response genes transcript frontloading. figshare. Dataset. https://doi.org/10.6084/m9.figshare.25515379.v1

Posadas, Niño; Da-Anoy, Jeric; Conaco, Cecilia (2024). Sequence similarity, ortholog analysis, and gene content comparison. figshare. Dataset. https://doi.org/10.6084/m9.figshare.25506859.v1

Posadas, Niño; Da-Anoy, Jeric; Conaco, Cecilia (2024). Differentially expressed genes. figshare. Dataset. https://doi.org/10.6084/m9.figshare.25389190.v1

Posadas, Niño; Da-Anoy, Jeric; Conaco, Cecilia (2024). Mapping files. figshare. Dataset. https://doi.org/10.6084/m9.figshare.25389169.v1

Posadas, Niño; Da-Anoy, Jeric; Conaco, Cecilia (2024). Annotation. figshare. Dataset. https://doi.org/10.6084/m9.figshare.25388845.v1

Posadas, Niño; Da-Anoy, Jeric; Conaco, Cecilia (2024). Predicted peptides. figshare. Dataset. https://doi.org/10.6084/m9.figshare.25388818.v1

Posadas, Niño; Da-Anoy, Jeric; Conaco, Cecilia (2024). Reference transcriptomes. figshare. Dataset. https://doi.org/10.6084/m9.figshare.25379302.v1

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
