# Peer review of "Interspecies differences in the transcriptome response of corals to acute heat stress"

_PeerJ, doi:10.7717/peerj.18627_

## Round 0.1 · original submission · Major Revisions

Two expert reviewers have evaluated your manuscript and their comments can be seen below and in an attached PDF. While both reviewers have positive comments about your study, they also have some important and insightful observations that I agree need to be addressed in a revised version of your manuscript. Please ensure that you address all of the comments in a rebuttal that clearly indicates where the changes of modification were made or else justify as to why not.

Reviewer 1 ·

Basic reporting

In this study, Da-Anoy and colleagues investigated the gene expression responses of four coral species to acute thermal stress. Corals were exposed to a thermal challenge and sampled at 4 and 24 hours along with untreated controls. RNA was extracted and sequenced for de novo transcriptome assemblies for each species. Using these transcriptomes, the authors conducted ortholog analyses in order to determine whether there had been gene expansion events in specific lineages in addition to identifying protein domains and known cnidarian stress response genes in the focal species and families. Additionally, they conducted a differential expression analysis and explored the gene expression plasticity of the four focal species to thermal stress. The authors show interesting differences in numbers of stress-related genes in different coral species, supporting the results of earlier studies. Additionally, they find that the most susceptible species in this study, S. caliendrium, has a significantly more plastic transcriptome response to thermal stress, and that more tolerant species showed evidence of frontloading of stress response genes under ambient temperatures.

I think that this manuscript is overall well-written and shows interesting results regarding the genetic underpinnings of interspecific differences in thermal tolerance in corals. Although the writing is overall quite clear, I think some sections require some revision and reorganizing in order to make the author’s points easier to follow – see my line-by-line comments in the attached document below.

Overall, I think the introduction introduces the importance of the study well and the authors do a good job reviewing the relevant literature. However, I think it overall needs some revision for English grammar to ensure the points being raised can be followed by readers. Additionally, I think it would be easier to follow groupings of corals into families later on in the results/discussion if you give a brief overview in the introduction of which families these species belong to, especially for readers less familiar with these species.

Experimental design

My main concern is about the sequencing of the species having been done on two different platforms. Although Dixon et al. 2020 (https://doi.org/10.1111/mec.15535) was a meta-analysis in which different BioProjects were mapped to the same reference genome, rather than de novo assembly, as in this current study, I think they raise an interesting point in that varying sequencing methods can have different outcomes. Could you include some supplementary figures and statistics which show overall patterns in the two sequencing platforms? Since two of the species were sequenced on HiSeq 4000, and the other on HiSeq 2500, I would want to see some evidence that the species differences outlined here cannot also be explained by differences caused due to the use of different sequencing runs/platforms.

For example, could you indicate in your supplemental tables showing the assembly statistics (perhaps via color-coding) which species were sequenced on which platform so that readers can more easily determine any trends? Are there any significant differences in the assembly statistics based on the sequencing platform used?

I would also be interested in seeing a PCA of all species/treatment samples together, colored/coded by sequencing platform to see if any trends emerge.

Validity of the findings

I think the discussion is well written and the authors place the results in context nicely. However, I found there was a slight mismatch between what is presented as the “main idea” in the beginning of the discussion and the focus of the results/figures/remaining discussion – differences in stress-related genes/gene family expansion in different coral species. I think this point could be underscored more clearly in the first paragraph of the discussion.

Annotated reviews are not available for download in order to protect the identity of reviewers who chose to remain anonymous.

·

Basic reporting

No comment

Experimental design

no comment

Validity of the findings

no comment

Additional comments

In this manuscript, the authors investigate the transcriptomic response to acute heat stress in four species of stony corals. The study is well-structured, with a clear and straightforward experimental design, and the results are rigorously described and analyzed. The discussion is framed within the context of current knowledge, providing valuable insights that better our understanding of coral responses to acute heat stress. Publishing this work would be highly beneficial to researchers studying the cellular and physiological responses of organisms to heat stress and would serve as a valuable resource for future projects and experiments.
I have the following questions and comments, as they appear in text:

L107. Could authors specify the location of the Bolinao-Anda Reef Complex?

L122. From the text, it appears the corals were placed directly from the acclimation tanks to the tanks with high-temperature water without any temperature ramping. From my experience, such a transition causes immediate shock to corals, and they usually die (even days later) with visible signs of tissue loss and necrosis, not through “regular” coral bleaching. Why did the authors decide not to go through temperature ramping, and could they discuss the possible effects of such immediate stress on the stress pathways that may differ from the more commonly used temperature stress approach through slower temperature ramping?

L205-210 Could authors describe more in detail how they identified SRGs, and gene regulatory elements in the coral transcriptomic dataset?

L223-224 Log2 fold change of 4 and higher is a very strict parameter to consider genes to be differentially expressed. I am more used to seeing log2 fold change of 2 and higher to be applied. Why did the authors decide to consider only genes with >4 change? I can imagine that applying a >2 threshold would change the results significantly.

L407-410. This is probably only a linguistic comment but I don’t fully understand the use of the word “therefore” in the second sentence. How does the information in the 1st sentence imply that 53 – 64 % of SRGs in these corals may be considered frontloaded?

L432 Why is log2 fold change >2 considered as “drastic upregulation” in this analysis, while in the methodology, the authors consider a significant change in gene expression log2 fold change of 4 and higher?

L432 I believe there is a typo in the Figure number and it should be Fig. 4C instead of 4D. Figure 4C is not discussed anywhere else in the results.

L528-531. Did the authors consider that the increase in DNA damage repair genes in heat-sensitive S. caliendrum may be caused by a not-that-effective antioxidant defense system in the given coral, compared to the other three corals? I believe the data in Figure 4C might corroborate this theory but I have not analyzed the data, it comes simply from looking at the figure.

General comment about the discussion:
This paper has an extensive dataset and the authors present many various analyses to discern every possible aspect of coral response to acute heat stress and how it differs between four different species. From this point of view, I find that the discussion does not fully reflect the scale of the experiment. The authors focus primarily on gene frontloading and DNA repair pathways, leaving all the other data in Figures 1, 3, and 4 (C) largely undiscussed. However, I entirely respect that it is the authors’ choice how they want to discuss their data, so please take this just as a comment.

---

## Round 0.2 · accepted · Accept

I am satisfied with the changes that have been made to the revised version of the manuscript.

·

Basic reporting

no comment

Experimental design

no comment

Validity of the findings

no comment

Additional comments

The authors answered all comments and suggestions satisfactorily; therefore, I endorse this manuscript for publication.